# The Association between the rs3747406 Polymorphism in the Glucocorticoid-Induced Leucine Zipper Gene and Sepsis Survivals Depends on the SOFA Score

**DOI:** 10.3390/ijms25073871

**Published:** 2024-03-30

**Authors:** Stefan Rusev, Patrick Thon, Tim Rahmel, Dominik Ziehe, Britta Marko, Hartmuth Nowak, Björn Ellger, Ulrich Limper, Elke Schwier, Dietrich Henzler, Stefan Felix Ehrentraut, Lars Bergmann, Matthias Unterberg, Michael Adamzik, Björn Koos, Katharina Rump

**Affiliations:** 1Klinik für Anästhesiologie, Intensivmedizin und Schmerztherapie, Universitätsklinikum Knappschaftskrankenhaus Bochum, 44892 Bochum, Germany; stefan.rusev@edu.ruhr-uni-bochum.de (S.R.); patrick.thon@rub.de (P.T.); tim.rahmel@rub.de (T.R.); dominik.ziehe@rub.de (D.Z.); britta.marko@kk-bochum.de (B.M.); hartmuth.nowak@kk-bochum.de (H.N.); lars.bergmann@kk-bochum.de (L.B.); matthias.unterberg@kk-bochum.de (M.U.); michael.adamzik@kk-bochum.de (M.A.); bjoern.koos@rub.de (B.K.); 2Center for Artificial Intelligence, Medical Informatics and Data Science, University Hospital Knappschaftskrankenhaus Bochum, 44892 Bochum, Germany; 3Klinik für Anästhesiologie, Intensivmedizin und Schmerztherapie, Klinikum Westfalen, 44309 Dortmund, Germany; bjoern.ellger@klinikum-westfalen.de; 4Department of Anesthesiology and Operative Intensive Care Medicine, Cologne Merheim Medical School, University of Witten/Herdecke, 51109 Cologne, Germany; limperu@kliniken-koeln.de; 5Department of Anesthesiology, Surgical Intensive Care, Emergency and Pain Medicine, Ruhr-University Bochum, Klinikum Herford, 32049 Herford, Germany; elke.schwier@klinikum-herford.de (E.S.); mail@d-henzler.de (D.H.); 6Klinik für Anästhesiologie und Operative Intensivmedizin, Universitätsklinikum Bonn, 53127 Bonn, Germany; stefan.ehrentraut@ukbonn.de

**Keywords:** glucocorticoid-induced leucine zipper, GILZ, single-nucleotide polymorphism, sepsis, personalized medicine, hydrocortisone, rs3747406

## Abstract

The variability in mortality in sepsis could be a consequence of genetic variability. The glucocorticoid system and the intermediate TSC22D3 gene product—glucocorticoid-induced leucine zipper—are clinically relevant in sepsis, which is why this study aimed to clarify whether TSC22D3 gene polymorphisms contribute to the variance in sepsis mortality. Blood samples for DNA extraction were obtained from 455 patients with a sepsis diagnosis according to the Sepsis-III criteria and from 73 control subjects. A SNP TaqMan assay was used to detect single-nucleotide polymorphisms (SNPs) in the TSC22D3 gene. Statistical and graphical analyses were performed using the SPSS Statistics and GraphPad Prism software. C-allele carriers of rs3747406 have a 2.07-fold higher mortality rate when the sequential organ failure assessment (SOFA) score is higher than eight. In a multivariate COX regression model, the SNP rs3747406 with a SOFA score ≥ 8 was found to be an independent risk factor for 30-day survival in sepsis. The HR was calculated to be 2.12, with a *p*-value of 0.011. The wild-type allele was present in four out of six SNPs in our cohort. The promoter of TSC22D3 was found to be highly conserved. However, we discovered that the C-allele of rs3747406 poses a risk for sepsis mortality for SOFA Scores higher than 6.

## 1. Introduction 

Sepsis is a life-threatening syndrome associated with organ dysfunction caused by a misled immunologic response to an infectious agent [1]. In a clinical setting, organ dysfunction is indicated by an increase in the Sequential Organ Failure Assessment (SOFA) score of 2 points or more [1]. The first indication is often a significantly reduced general condition characterized by non-specific symptoms such as chills, hyperventilation, decreased vigilance, tachycardia, and hypotension [2]. Sepsis is, therefore, a serious illness and usually requires treatment in the intensive care unit [3]. In most cases, sepsis is triggered by a bacterial infection. Fungi, viruses, or parasites are less frequently responsible. The most common pathogens include E. coli, streptococci, staphylococci, pseudomonads, bacilli, enterococci, *Enterobacter* spp., *Candida* spp., and Klebsiella [4]. It is accompanied by an incidence of 189 hospital-treated adult cases per 100,000 person years and a mortality rate of 26.7% worldwide [5]. The mortality rate varies considerably between different patient groups due to a large number of known and unknown factors. These factors include health status, concomitant medical treatment, and the constitution of the human genome, which characterizes various physiological and biochemical systems [6,7]. In this sense, there is an urgent need to clarify genetic factors that may influence the clinical outcome of sepsis. Furthermore, evidence of the unexplored causalities for variation in sepsis mortality targets patient enrichment so that only a certain group of patients may benefit from a distinct therapy [8,9,10]. Patient enrichment can be generally defined as a design feature or strategy by which patients who meet eligibility criteria are allocated into different treatment arms or study cohorts. Genetic variants may be an important factor in narrowing down these study cohorts. An interesting candidate gene could be TSC22D3, which codes for GILZ—the glucocorticoid-induced leucine zipper. GILZ is an important mediator of the glucocorticoid system, which is physiologically and biochemically important in sepsis. Respectively, both GILZ and the glucocorticoid system are clinically relevant in sepsis and regulate the adaptive and innate axis of the immune system [11,12]. The glucocorticoid system can be therapeutically influenced by glucocorticoids, which are frequently prescribed in the treatment of sepsis with variable success [13,14,15]. Current guidelines recommend glucocorticoid treatment for patients with persistent shock who require vasopressors, as there is evidence of faster shock reversal and reduced vasopressor dependency. A recent meta-analysis suggests a potential benefit of glucocorticoid treatment in a subset of patients with septic shock who are severely ill and have a pulmonary infection, but further research is needed to validate these findings and better understand the differences within sepsis patients [16]. The GILZ, as a crucial mediator of the glucocorticoid system, promotes the inhibitory effect on the pro-inflammation and stimulates the anti-inflammation as a blocker of pro-inflammatory cascades, such as the NF-kB and MAPK pathways [17,18,19]. Through these pathways, the GILZ affects cells of adaptive and innate immunity [20,21,22,23], which, in turn, influence the pathogenesis and progression of sepsis [24,25]. Hence, there is a clear indication that the GILZ might have a pivotal role in the development and clinical outcome of sepsis. The aim of this study was, therefore, to investigate the following hypotheses: (1) genetic variations in the TSC22D3 gene are detectable in a cohort of sepsis and control subjects, and (2) the genetic variants detected have an impact on the outcome of sepsis. 

## 2. Results

### 2.1. Patients’ Characteristics

Samples from 455 patients diagnosed with sepsis and 73 control patients with abdominal surgery but without sepsis were analyzed. A total of 455 patients were genotyped for the assessed SNPs, and 40 patients were excluded from survival analysis due to missing values for 30-day mortality. The majority of septic patients were male (64.9%), the median SOFA Score was 8.5, the 30-day mortality was 30.1%, and the focus of infection was on the lower respiratory tract (Table 1). 

The majority of control subjects were male (52.1%), the median age was 64 years, and the median SOFA score was 3 (Table 2).

### 2.2. Polymorphism Analysis in the Cohort

We tested our first hypothesis and searched for six single-nucleotide polymorphisms (SNPs) in the GILZ gene sequence in the genome of septic patients and in the genome of our control subjects. Although they are the most frequent and have a reasonable global allele frequency, four of the six SNPs (rs3924026, rs4300127, rs73525022, and rs4342758) had only the wild-type allele present in the cohort. Only two SNPs (rs3747406 and rs17254207) had both alleles present in the analyzed individuals (Table 3). There was no difference between sepsis and control patients regarding the allele frequencies of both alleles of the tested polymorphisms (*p* = n.s.).

### 2.3. Impact of Polymorphisms on Sepsis Survival

To assess the effect of the genotype on survival, we focused only on the SNPs where the minor allele was present in the patient cohort studied: rs17254207 and rs3747406. We investigated whether the polymorphisms correlated with sepsis survival. The rs17254207 polymorphism showed no correlation with sepsis survival (r = 0.013; *p* = 0.838). In contrast, the rs3747406 polymorphism showed a correlation with the 30-day sepsis mortality in critically ill sepsis patients. The Kaplan–Meyer analysis showed that the negative impact of the C-allele on survival increases with a rising SOFA score (Figure 1a–c). C-allele carriers with a SOFA score ≥ 6 have a 1.90-fold higher mortality rate (Figure 1b), whereas a SOFA score ≥ 8 leads to a 2.03-fold higher mortality rate within the C-allele carriers (Figure 1c). 

Univariate and multivariate COX regression was performed to analyze the impact of various factors on survival. Interestingly, the rs3747406 polymorphism was the strongest prognostic factor in a multivariate model, with a hazard ratio of 2.1 and independent of age and gender.

The rs3747406SNP with a SOFA score ≥ 8 was found to be an independent risk factor for 30-day mortality in sepsis in a multivariate Cox regression model (HR = 2.12, *p* = 0.0011, CI (1.193; 3.796)); (Table 4). The C-allele was an independent risk factor for 30-day mortality in sepsis in critically ill patients. C-allele carriers with a SOFA score ≥ 8 had a 2.12-fold higher mortality rate in sepsis. 

### 2.4. The GILZ mRNA Expression Analysis in Septic Patients

To elucidate the possible underlying mechanism for the presence of the SNP rs3747406 C-allele, GILZ mRNA expression was analyzed depending on genotype. Interestingly, GILZ mRNA expression was decreased in TT genotypes (*p* = 0.0059; Figure 2) but increased in C-allele carriers over the time course of sepsis from day 1 to day 8 (*p* = 0.0093; Figure 2). 

## 3. Discussion

This is the first study examining polymorphism in the TSC22D3 gene in septic patients. After database analysis, we investigated six polymorphisms within the TSC22D3 gene (NCBI.snp. and thermofisher.org, both accessed on 26 January 2024) [26,27]. The TSC22D3 gene sequence is highly conserved in a European cohort. The cohort analyzed here had only the wild-type allele in four out of six polymorphisms, and there were no differences in the allele distribution between the sepsis and the control group. Only rs3747406 and rs17254207 showed both alleles with low frequencies of the minor alleles. 

In a further analysis, we examined whether these minor alleles might have an impact on sepsis survival. In fact, a correlation between rs3747406 and sepsis survival was demonstrated in sepsis patients with a SOFA score ≥ 6. 

To date, little has been published about polymorphisms in the TSC22D3 gene, and our data suggest that the TSC22D3 gene is highly conserved in European populations. TSC22D3 belongs to the evolutionary conserved TSC-22 domain family, which is associated with the transforming growth factor-β 1 (TGF-β1)-stimulated clone 22 (TSC22) family, which compromises proteins with a leucine zipper domain and a TSC-box [28,29,30]. A limited amount of data is currently available on the function of this protein family, especially regarding the TSC-22 homologous gene-1 (THG-1)/TSC22 domain family member 4 (TSC22D4) [31]. The TSC-22 domain family includes four members: TSC22D1/TSC-22, TSC22D2/KIAA0669, TSC22D3/GILZ, and TSC22D4/THG-1. The best-characterized member of this protein family is TSC22D1, which possesses a tumor-suppressor function, and TGF-β1-stimulated clone 22 (TSC-22/TSC22D1) was first identified as a target gene of TGF-β1 in mouse osteosarcoma cells [32,33]. In addition, our studies indicated a low frequency of TSC-22 SNP alleles in the population [34]. Hence, the low frequency of mutant alleles in our analysis is consistent with published data.

We are able to show that the rs3747406 polymorphism influences sepsis survival in critically ill patients and GILZ expression over the time course of sepsis. Thus, the presence of this SNP rs3747406 in the genome can be considered as a possible explanation for the variance in sepsis survival. Although the exact mechanism of C-allele activity remains elusive, we have shown that C-allele carriers have a 2-fold higher GILZ mRNA expression on day 8 than on day 1 of sepsis diagnosis and a 3-fold higher expression than T-allele carriers on day 8. Hence, polymorphism appears to affect GILZ expression over the course of sepsis. In this context, it should be discussed that the effect of the polymorphism rs3747406 in a multivariate analysis depends on the SOFA score and, to a lesser extent, on age. The SNP rs3747406 is located near the 5′-untranslated region of the GILZ-encoding mRNA [35]. Previous studies have shown that different 5′-untranslated regions can lead to the expression of altered isoforms of the same mRNA [36,37,38]. Remarkably, it has been reported that both the leucine zipper and the C-terminus rather than the N-terminus of the GILZ amino acid sequence are essential for homodimerization and NF-kB arrest in the cytosol, respectively [39]. Therefore, it can be hypothesized that the C-allele of the rs3747406 may contribute to an impaired structure of the GILZ protein, which in turn may lead to inadequate regulation of the immune system via the GILZ, aggravating the immunopathology of sepsis [40]. Another explanation for the effect of the SNP could be that the polymorphism alters the binding of transcription factors [41,42]. Since patients with higher SOFA Score have an altered concentration of transcription factors (e.g., EZH2), an altered binding of transcription factors could explain the SOFA Score-dependent effect of the SNP [43]. 

A further explanation for both observations that C-allele carriers have a higher mortality rate depending on the SOFA Score and a higher GILZ expression is the finding that the GILZ is responsible for reduced responsiveness of monocytes to lipopolysaccharides [44]. A crucial influence of the GILZ is the observed inhibition of IFN-y production, which in turn is essential for the initiation of pro-inflammatory M1 macrophages [45,46]. Excessively reduced reactivity of monocytes and reduced differentiation to pro-inflammatory M1 macrophages may lead to reduced bacterial clearance in the initial phase of sepsis disease [47,48]. This may also explain why the effects of the C-allele correlate with the SOFA Score—the higher the risk of death, the higher the SOFA Score and the higher the need for a well-proportioned and synchronized host immune response to the infectious agent [49,50]. 

Our data address an important clinical question about the outcome of sepsis. One question that has not yet been adequately addressed is the reason for the variation in mortality rates in sepsis, which emphasizes the importance of further research [51,52]. Therefore, there is an urgent need to identify the factors responsible for the different mortality rates in sepsis. In order to find candidate genes that can shed light on the variability in sepsis mortality, this study focused on the TSC22D3 gene and its product GILZ as one of the key glucocorticoid mediators, which has been considered a promising protein for sepsis survival in mouse models [53,54]. This study focused on the investigation of polymorphisms in the GILZ gene, their presence in septic patients and control subjects, and their impact on the sepsis mortality rate. We chose surgical patients as a control group, as we aimed to include patients with comparable risk for sepsis development. If we had seen differences in the occurrence of one variant, either the sepsis or the control group, this variant could possibly make one group more prone to sepsis development. However, we did not see any differences in genotype distribution between the control group and septic patients. Hence, the anti-inflammatory reaction in sepsis patients could be altered, which impacts pathogenesis. The effect of the rs3747406 SNP was dependent on the SOFA Score but independent of age and gender [35,55]. The data presented show that different polymorphisms in the human genome are an essential component for understanding the different mortality rates in sepsis. For example, the deletion allele of the NFκB1 insertion-deletion (-94ins/delATTG) polymorphism has been associated with an increased 30-day mortality rate in septic patients [56]. As described above, the GILZ is a crucial mediator of the glucocorticoid system, which, in turn, is of clinical importance in sepsis [12,57]. Thus, the hypothesis that the rs3747406 polymorphism is a possible marker to discriminate sepsis patients who benefit from corticosteroid therapy needs to be tested in further studies. It can be speculated that the polymorphism can alter GILZ expression after glucocorticoid treatment. GILZ expression is susceptible to induction by glucocorticoids. Dexamethasone can increase GILZ transcripts more than tenfold in human RA synovial fibroblasts at concentrations as low as 1 nM. Furthermore, GILZ expression decreases when circulating cortisol is reduced in humans [58]. Hence, as the polymorphism is in a regulatory region of the TSC22D3 gene, it might alter responsiveness to glucocorticoids, e.g., via glucocorticoid receptor binding in septic patients. This question is further emphasized by the observations of Schäfer and colleagues that the presence of a specific NF-kB1 promoter polymorphism would lead to an association of hydrocortisone with a higher mortality rate in sepsis, prompting clinicians to consider patient groups with different mortality rates [59]. Our data and the specific factors previously identified could empower clinicians to differentiate patient groups with distinct mortality rates, essentially allowing precision medicine [8,60]. Individualized treatment will play a central role in medical guidelines, with increasing evidence that hydrocortisone is not suitable for every patient with sepsis. For example, Antcliffe and colleagues showed that two transcriptomic configurations can be distinguished in septic patients, one of which has an increased mortality rate when receiving corticosteroids [9]. In addition, König and colleagues showed that the serum IFNγ/IL10 ratio can be used to predict survival in patients receiving hydrocortisone [10]. There are also several biomarkers that may serve as potential parameters for predicting the efficacy of corticosteroids in sepsis. For example, Bentzer and colleagues demonstrated that corticosteroid treatment increases 28-day survival when plasma cytokine levels of CCL4, IL3, and IL6A are elevated and decreases when the levels of these biomarkers are low [61]. 

A limiting aspect for the interpretation of our results is the number of severely ill patients whose genome is analyzed for the rs3747406 polymorphism. Samples from patients of other ethnicities need to be analyzed for this genotype to determine decisively whether the C-allele is a marker for patient enrichment. We calculated the expected allele distribution for rs3747406 with the European allele frequency C = 0.17, which differs slightly from the global distribution of C = 0.22. Of note, the frequency of rs3747406 polymorphisms in the African population is C = 0.47; thus, further research is required to assess the impact of the polymorphism on sepsis mortality in ethnicities other than Europeans. In addition, further research should be conducted to determine how the mechanism of action affects the 30-day survival rate in sepsis when C-allele carriers of SNP rs3747406 have increased quantitative GILZ mRNA. For this reason, further studies are essential. Another limitation that has to be mentioned is the relatively high number of missing values in the baseline characteristics, which is due to the multicentric approach of our study and has to be optimized in future research. 

## 4. Materials and Methods

### 4.1. Patient Recruitment

Enlistment of sepsis patients to the SepsisDataNet.NRW project and controls were approved by the Ethics Committee of the medical faculty of the Ruhr University of Bochum (No. 18-6606—BR) and by the Ethics Committees of the University of Münster (No. 2017-513-b-S) and the University of Bonn and accomplished if the criteria of the Sepsis-III definition were fulfilled by the sepsis patients [1]. Patients were recruited at the Knappschaftskrankenhaus Bochum GmbH (University of Bochum), St. Elisabeth Gruppe GmbH, and Herford (University of Bochum), as well as at the Clinics of Anesthesiology (University of Münster and University of Bonn). In the SepsisDataNet.NRW, peripheral blood was collected within the first 36 h after diagnosis of sepsis (day one), day four, and day eight after study inclusion. Informed consent was given by all patients or their representatives. Control subjects were defined as patients who had undergone surgery in the abdominal cavity and did not have sepsis according to the Sepsis-III definition [1].

### 4.2. Blood Sample Collection, Preparation, and Storage

Whole blood was collected from control subjects and patients who fulfilled the criteria of sepsis according to the Sepsis III definition [1]. The DNA-Exact tube (Sarstedt, Nümbrecht, Germany) and RNA Tube (Applied Biosystems, Life Technologies, Darmstadt, Germany) were used for DNA and RNA blood extraction, respectively. Total DNA and RNA were extracted from whole blood samples using the QIAamp and RNeasy kits, respectively, according to the manufacturer’s instructions (QIAGEN, Hilden, Germany). DNA and RNA sample aliquots were stored at −80 °C until further analysis.

### 4.3. SNP TaqMan Assay

In order to examine the SNPs rs3747406, rs3924026, rs4300127, rs73525022, rs4342758 and rs17254207 in samples from 455 sepsis patients and 73 control probands, SNP TaqMan assay (Thermo Fisher Scientific, Darmstadt, Germany) was utilized for standard qPCR (CFX Connect Real-Time System, Bio-Rad Labs, Hercules, CA, USA) using the TaqMan Genotyping Master Mix (Thermo Fisher Scientific, Darmstadt, Germany). The selection of the SNPs was based on a review of the NCBI database. A total of 24 µL of the final SNP reaction mix (consisting of 12.5 µL Master Mix, 1.25 µL SNP TaqMan Assay Primer, 10.25 µL nuclease-free water) and 1 µL of DNA (10 ng/µL) were dispensed in an optical reaction plate. RT-PCR amplification was performed under the following conditions: one denaturation step at 95 °C for 10 min, followed by 40 cycles at 95 °C for 15 s and at 60 °C for 1 min. Each genotype was assigned after the fluorescence emission of every sample was recorded at the VIC and FAM dye wavelengths. 

### 4.4. GILZ mRNA Quantification

The RT-qPCR was performed in duplicate using the GoTaq qPCR Master Mix (Promega, Madison, WI, USA) and the subsequent primers on a CFX Connect Real-Time System (Bio-Rad Labs, Hercules, CA, USA). A total 25 µL reaction mix (consisting of 12.5 µL GoTaq qPCR Master Mix, 2.5 µL cDNA (10 ng/µL), 1 µL sense primer, 1 µL anti-sense primer, 8 µL nuclease-free water) was distributed in an optical reaction plate. RT-qPCR amplification was carried out under the following conditions: a primary denaturation step at 95 °C for 2 min, followed by 40 cycles at 95 °C for 30 s and at 60 °C for 30 s. The following primers were utilized: TSC22D3_mRNA—forward primer sequence 5′->3′ GTTAAGCTGGACAACAGTGCCT, reverse primer sequence 5′->3′ TTCTCCACCAGCTCTCGGAT (Eurofins Scientific SE, Hamburg, Germany). The relative GILZ mRNA expression was standardized to the housekeeping gene ß-actin, forward primer sequence 5′->3′ CCTTCCTGGGCATGGAGT, reverse primer sequence 5′->3′ CAGGGCAGTGATCTCCTTCT (Eurofins Scientific SE, Hamburg, Germany) and calculated using the 2^−∆∆CT^ method. 

### 4.5. Statistical Analyses

Characteristics of the individuals involved are described as numbers and percentages for categorical variables, means and standard deviations (±SD), or medians with assigned interquartile ranges for continuous variables. Kaplan–Meier survival curves were generated to evaluate the survival rate of the septic groups divided by the presence of the SNP rs3747406 genotype. The Log Rank test was used to determine the statistical significance of the survival effect observed. The hazard ratio was determined using univariate Cox Regression analysis. We evaluated the independence of the SNP rs3747406 on survival from the SOFA score, age, gender, and hydrocortisone administration using a multivariate Cox Regression analysis. Univariate and multivariate analyses were used to calculate hazard ratios with corresponding 95% confidence intervals to approximate the extent of associations between covariates and the time of death. The Hardy–Weinberg equilibrium was assessed for the polymorphism, and Chi quadrat values were calculated for the variation in allele distribution of each polymorphism between the sepsis and control groups. *p* values of less than 0.05 were considered statistically significant. All statistical analyses were performed using SPSS Statistics (Version 28.0.0.0, IBM, Armonk, NY, USA).

## 5. Conclusions

This study has first shown that the GILZ gene is highly conserved. However, this study identified the SNP rs3747406 in the TSC22D3 gene encoding the GILZ as a potential marker for predicting 30-day mortality in sepsis, depending on the SOFA Score. This SNP is, therefore, a conceivable factor in explaining the different mortality rates. Further studies are needed to verify whether the SNP rs3747406 can be used as a potential clinical indicator for the prediction of 30-day survival in sepsis and whether there is a survival effect of increased GILZ expression in terms of personalized medicine.

## Figures and Tables

**Figure 1 ijms-25-03871-f001:**
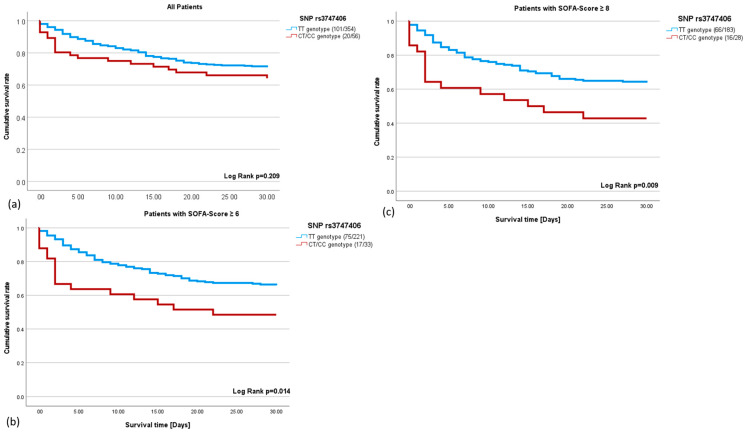
(**a**–**c**): The 30-day survival rate in sepsis using the Kaplan–Meier analysis. (**a**). The survival curve of the T-allele carriers (blue) and C-allele carriers (red) for the single nucleotide polymorphism (SNP) rs3747406 (HR = 1.36; *p* = 0.215). (**b**): Septic patients with a sequential organ failure assessment (SOFA) score ≥ 6 and T-allele (blue), and septic patients with a SOFA score ≥ 6 and C-allele (red) for the SNP rs3747406 (HR = 1.90; *p* = 0.017). (**c**): Septic patients with a SOFA score ≥ 8 and T-allele (blue), and septic patients with a SOFA score ≥ 8 and C-allele (red) for the SNP rs3747406 (HR = 2.027; *p* = 0.011).

**Figure 2 ijms-25-03871-f002:**
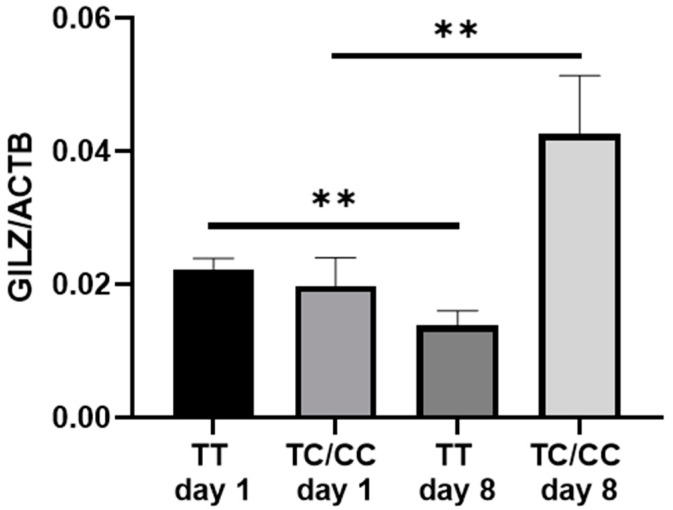
Quantitative glucocorticoid-induced leucine zip (GILZ) mRNA expression in all rs3747406 SNP genotypes on day 1 and day 8 of sepsis diagnosis. The quantitative gene expression was normalized to the housekeeping gene ß-actin, computed using the 2^−∆CT^ method. TT day 1 to day 8 *n* = 87; ** *p* = 0.0059; TC/CC day 1 to day 8 *n* = 13, ** *p* = 0.0093.

**Table 1 ijms-25-03871-t001:** Baseline characteristics of sepsis patients.

	Total Cohort	*n* = 415
Sex male *n* (%)	242 (64.9)	373
Age in years median [IQR]	65 [55–76]	373
SOFA Score median—day 1 [IQR]	8.5 [5–12]	329
PCT—day 1(ng/mL)	10.24 ± 20.02	115
CRP—day 1 (mg/dL)	17.9 ± 11.47	188
ICU stay (days)	9 [3–17]	268
30-day mortality (%)	125 (30.1)	415
Survival time days median [IQR]	30 [15–30]	410
Infection focus *n* (%)		110
– CNS	2 (1.8)	
– Lower respiratory tract	56 (50.9)	
– Skin and soft tissues	7 (6.4)	
– Urogenital tract	12 (10.9)	
– Cardiovascular	7 (6.4)	
– Intra-abdominal	24 (21.8)	
– Musculoskeletal	2 (1.8)	

Missing values: sex (male)—42; Age—48; SOFA score—86; PCT day 1—300, CRP day 1—227; ICU stay—147; unknown infection focus—305.

**Table 2 ijms-25-03871-t002:** Baseline characteristics of control patients.

	Total Cohort	*n* = 73
Sex male *n* (%)	38 (52.1%)	73
Age in years median	64 [56–76]	73
SOFA Score median—day 1	3 [1–6]	52
Comorbidities (%)		73
– Alcohol abuse	8 (11)	
– Lung disease	11 (15.1)	
– Hypertension	46 (63)	
– Chronic kidney disease	4 (5.5)	
– Chronic obstructive pulmonary disease	12 (16.4)	
– Diabetes	9 (12.3)	
– Obesity	15 (20.5)	
– Cardiovascular	23 (31.5)	
– Malignant Tumor	58 (79.5)	
– Nicotine Abuse	26 (35.6)	
– Transplantation	1 (1.4)	

**Table 3 ijms-25-03871-t003:** Single-nucleotide polymorphisms (SNPs) were examined, including SNP type, global allele frequency, and the total number of subjects with the corresponding genotype.

Single-Nucleotide Polymorphism (SNP)	Kind of Polymorphism	SNP Allele Frequency, Global According to NCBI.SNP	Genotype	Patients with Sepsis	Control Probands	Chi Quadrat X² (Variation in the Allele Distribution between Sepsis and Control Subjects)
rs3747406	C/T Transition Substitution	C (0.17)	TT	391 (85.9%)	62 (86.1%)	*p* = 0.690
			TC	29 (6.4%)	6 (9.6%)	
			CC	35 (7.7%)	4 (6.4%)	
rs3924026	A/G Transition Substitution	G (0.09)	AA	192 (100%)	29 (100%)	No variation
			AG	0 (0%)	0 (0%)	
			GG	0 (0%)	0 (0%)	
rs4300127	A/G Transition Substitution	G (0.05)	AA	229 (100%)	29 (100%)	No variation
			AG	0 (0%)	0 (0%)	
			GG	0 (0%)	0 (0%)	
rs73525022	A/G Transition Substitution	G (0.03)	A	193 (100%)	29 (100%)	No variation
			AG	0 (0%)	0 (0%)	
			GG	0 (0%)	0 (0%)	
rs4342758	A/G Transition Substitution	G (<0.001)	AA	254 (100%)	12 (100%)	No variation
			AG	0 (0%)	0 (0%)	
			GG	0 (0%)	0 (0%)	
rs17254207	A/G Transition Substitution	G (0.11)	AA	184 (69.9%)	20 (68.9%)	*p* = 0.9547
			AG	40 (15.2%)	5 (17.2%)	
			GG	39 (14.8%)	4 (13.7%)	

**Table 4 ijms-25-03871-t004:** Both univariate and multivariate Cox regression were conducted to analyze the rs3747406 SNP in a sample of 211 patients with a SOFA score ≥ 8.

	Univariate Cox Regression				Multivariate Cox Regression			
	Hazard Ratio (HR)	95% Confidence Intervals for HR		*p*-Value	Hazard Ratio (HR)	95% Confidence Intervals for HR		*p*-Value
SNP rs3747406	2.027	1.173	3.502	0.011	2.128	1.193	3.796	0.011
Age	1.028	1.014	1.051	0.001	1.034	1.015	1.053	0.001
Sex (male)	1.160	0.727	1.850	0.533	1.067	0.663	1.715	0.790

## Data Availability

The data presented in this study are available on request from the corresponding author.

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
