# Peer review of "The Association between the rs3747406 Polymorphism in the Glucocorticoid-Induced Leucine Zipper Gene and Sepsis Survivals Depends on the SOFA Score"

_ijms, 2024, doi:10.3390/ijms25073871_

Round 1

Reviewer 1 Report

Comments and Suggestions for Authors

Dear Authors,

Please pay attention to the following questions and comments, pertaining to your manuscript:

1.      Do you believe that the aseptic inflammation after undergoing an abdominal surgery in the control group could have influenced your results? What was the reason of not choosing non-surgical patients as a control group? Please discuss in your text.

2.      Line 74. Overall 45 patients were excluded from the survival analysis due to missing values for SOFA Score. On the other hand, according to Table 1, there were 89 patients with missing SOFA score data (415-329). Please elucidate this point.

3.      Table 1. There were 42 (415-373) patients with no data about their sex and age, which were significant parameters for the multivariate analysis. What was the reason of not excluding these patients from the analysis? Please elucidate this point.

4.      Table 1. There is a lot of missing data, referring to laboratory parameters (PCT and CRP) and the infection focus. What is the reason of that? Please explain.

5.      Line 121. You probably mean Table 4 and not Table 5.

Best Regards

Author Response

Thank you very much for reviweing our manuscript. Please find the response in the file. 

Reviewer 2 Report

Comments and Suggestions for Authors

Rusev et al studied the effect of GILz gene polymorphisms on sepsis risk and sepsis outcome. They demonstrated that a particular polymorphism, rs3747406, correlates with the severity of sepsis and its outcome. The study is well structured and interesting for the reader. I only have a few suggestions.

The introduction should report the clinical characteristics of sepsis and the most frequent pathogens involved, as well as the therapeutic role of glucocorticoids

The discussion should explain how the trivate polymorphism can influence the response to glucocorticoids and the pathogenesis of sepsis.

Author Response

Thank you very much for reviweing our manuscript. Please find the reply in the attached file. 

Round 2

Reviewer 1 Report

Comments and Suggestions for Authors

Dear Authors,

thank you very much for providing comprehensive and convincing answers to my questions and queries, and made changes that have contributed to quality improvement and increased the publishing potential of your work. I have no further questions and queries, pertaining to your manuscript.

Best Regards